# Compartmentalization-induced phosphorescent emission enhancement and triplet energy transfer in aqueous medium

Zijian Li[1], Yifei Han[1] & Feng Wang [1]

Triplet energy transfer occurs frequently in natural photosynthetic organisms to protect against photo-oxidative stress. For artificial light-harvesting systems, several challenges need to be addressed to realize triplet energy transfer especially in aqueous medium. Specifically, the phosphors should be shielded from water and molecular oxygen, which facilitate to maintain intense emission intensity. Moreover, the donor–acceptor phosphors should be organized in close proximity, yet simultaneously avoiding direct homo- and hetero-interactions to minimize the potential energy losses. Herein an effective strategy has been developed to meet these requirements, by employing a rod–coil amphiphile as the compartmentalized agent. It renders synergistic rigidifying and hydrophobic shielding effects, giving rise to enhanced phosphorescent emission of the platinum(II) complexes in aqueous environment. More importantly, the donor–acceptor platinum(II) phosphors feature ordered spatial organization in the ternary co-assembled system, resulting in high light-harvesting efficiency. Therefore, the compartmentalization strategy represents an efficient approach toward color-tunable phosphorescent nanomaterials.

[1] CAS Key Laboratory of Soft Matter Chemistry, iChEM (Collaborative Innovation Center of Chemistry for Energy Materials), Department of Polymer Science and Engineering, University of Science and Technology of China, 230026 Hefei, Anhui, P. R. China. Correspondence and requests for materials should be addressed to F.W. (email: drfwang@ustc.edu.cn)

Light-harvesting system exists widely in natural photo-synthetic organisms, capable of funneling excitation energy from sun to the reaction center[1,2]. Nowadays, great efforts have been devoted to emulate nature's design to capture light energy, by bringing donor (D) antennae and acceptor (A) units together[3–5]. Despite the great progress achieved, the vast majority of current artificial harvesting systems are focused on singlet energy transfer (ET) between fluorescent chromophores[6–14]. The situation is a little different from natural photosynthetic organisms, in which triplet ET occurs frequently between D and A phosphors to protect against photo-oxidative stress[2]. For artificial light-harvesting antennae, the triplet excitons feature long excited-state lifetime and slow excitation energy release, which endow the systems with exceptional material properties[15–19].

Supramolecular D–A co-assembly is regarded as an efficient, modular approach toward artificial light harvesting in aqueous media[4]. It provides excellent water solubility and meanwhile avoids tedious multi-step synthesis. However, several challenges are met to achieve high triplet ET efficiency. First, the D/A phosphors should possess intense emission intensity in aqueous environment. A prerequisite for achieving this goal is to shield the phosphors from water and dissolved molecular oxygen, which facilitate to stabilize their triplet excitons[20,21]. Second, the D–A phosphors should be organized with close proximity, yet simultaneously avoiding direct homo- and hetero-interactions, in order to minimize the potential energy losses[22–24]. Owing to these restrictions, the previous triplet ET systems are mainly performed in crystalline and gel states[25,26], yet rarely explored in the aqueous media[27].

A feasible strategy to attain this objective is creating compartmentalization in supramolecular D–A systems[28–31], which avoids mutual interference and thereby improves ET efficiency. As a first embodiment of this design, herein aqueous light-harvesting systems are developed, with the involvement of two platinum(II) phosphors (1 and 2 serve as D and A, respectively, Fig. 1). In principle, these square planar platinum(II) phosphors suffer from severe excited-state deactivation in water (Supplementary Figs. 1 and 2). To address the issue, rod−coil amphiphile 3 (Fig. 1) is designed as the compartmentalized agent, possessing non-emissive gold(III) headgroup and hydrophilic polyethylene glycol 2000 (PEG2000) tail. 3 tends to self-assemble into micelles in water, providing hydrophobic interior to encapsulate platinum (II) complexes 1 (or 2) (Fig. 1). As a consequence, these platinum (II) emitters are shielded from the surrounding quenchers such as water and dissolved oxygen molecules, giving rise to phosphor-escent emission enhancement in aqueous environment. More importantly, the 1–3 (or 2–3) hetero-complexation prevails over the homo-complexation ones, ascribed to the strong π-stacking strength between neutral gold(III) headgroup on 3 and positively charged platinum(II) phosphors. For the ternary co-assembled system 1•2•3 (Fig. 1), the two platinum(II) phosphors are densely packed, yet separately partitioned via the compartmentalized agent 3. Highly efficient light harvesting and ET can be realized in aqueous medium, by taking advantage of triplet exciton stabilization and ordered spatial organization for D–A phosphors 1–2. Hence, the compartmentalization strategy exemplified in the current study provides a feasible approach toward color-tunable phosphorescent nanomaterials.

## Results

**Self-assembly behaviors of 3**. As an initial step, we study the self-assembly behaviors of rod−coil amphiphile 3. Depending on transmission electron microscopy (TEM) and dynamic light scattering (DLS) measurements, 3 is capable of forming micelles in water, with an averaged hydrodynamic diameter of 34 nm (Fig. 2a, b). The critical micelle concentration in water is deter-mined to be $6.7 \times 10^{-6}$ M via Nile red encapsulation experiment (Supplementary Fig. 3). In sharp contrast, no aggregates form for 3 in methanol under the same conditions (Fig. 2b and Supple-mentary Fig. 4). The reversible self-assembly of 3 is manifested via solvent-dependent ultraviolet–visible (UV-Vis) measure-ments. Upon varying the methanol/water volume ratio, an iso-sbestic point emerges at 407 nm for the metal-perturbed π-π* absorption (ranging from 355 to 420 nm, Fig. 2c). Notably, 3 is non-emissive in water ($\Phi_{em} < 0.001$, Supplementary Fig. 5), owing to the presence of thermally accessible d-d or LLCT (ligand-to-ligand charge transfer) states for the [Au(III)(C^N^C)(C≡C-R)] unit[32,33].

**Co-assembly behaviors between 1 and 3**. Co-assembly behaviors between 3 and the platinum(II) phosphor 1 are then investigated. Although 1 itself is insoluble in pure water, a transparent, yellow-colored solution emerges upon mixing 3 and 1 together (Fig. 3a). The enhanced solubility arises from the encapsulation of 1 into the inner micelle core of 3. Simultaneously, the typical metal-to-ligand charge transfer (MLCT)/LLCT absorbance of 1 [$\lambda_{max} = 420$ nm in water/MeOH (90:10, $v/v$)] undergoes bathochromic shift in complex 1•3 ($\lambda_{max} = 480$ nm, Fig. 3b). It originates from hetero π-π stacking between 1 and the [Au(III)(C^N^C)(C≡C-R)] unit on 3, which is supported by the close inter-planar dis-tance (3.44 Å) in density functional theory (DFT) theoretical

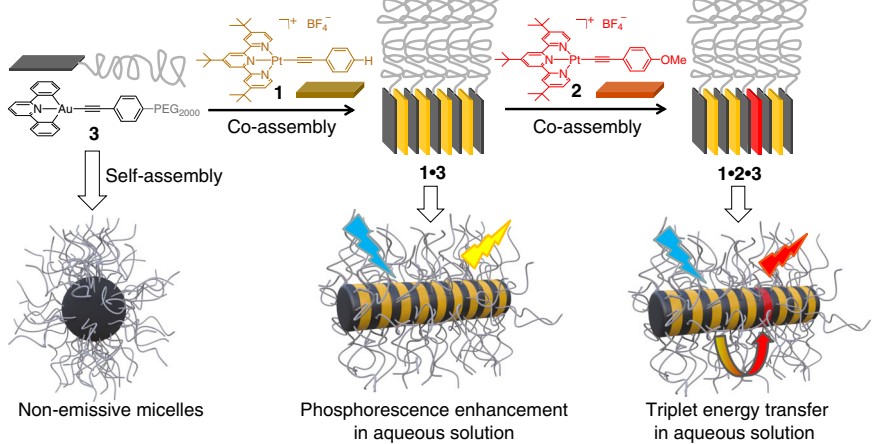

**Fig. 1** Molecular structures design and aqueous light-harvesting applications. Schematic representation for the phosphorescent enhancement and triplet energy transfer of platinum(II) complexes **1–2** in aqueous environment by employing the compartmentalized compound **3**

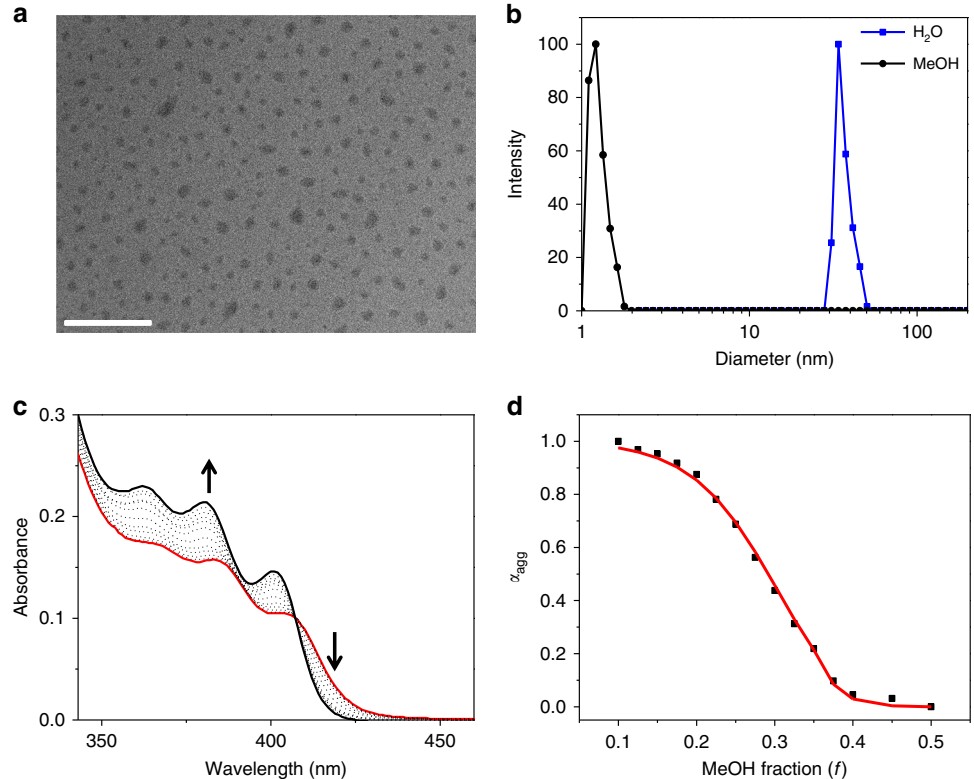

**Fig. 2** Characterization of the self-assembled system from **3**. **a** Transmission electron microscopic image recorded on a copper grid by drop-casting a $5.00 \times 10^{-5}$ M solution of **3** in water. Scale bar: 500 nm. **b** Hydrodynamic size distribution of a $5.00 \times 10^{-5}$ M solution of **3** measured by dynamic light scattering at 298 K. **c** Solvent-dependent ultraviolet–visible absorption spectra of **3** [$5.00 \times 10^{-5}$ M in water/MeOH (from 90:10 to 50:50, v/v)]. The arrows indicate the spectral change upon increasing the MeOH fraction. **d** Degree of aggregation ($\alpha_{agg}$) as a function of MeOH volume fraction ($f$) monitored at 420 nm. The red line denotes the mathematical fitting of the curve, according to the solvent-dependent equilibrium model (see Eqs. (1) and (2) in "Methods"). Accordingly, $\Delta G_{(f=0.1)}$ [denoting the Gibbs free energy gain upon monomer association in water/methanol (90:10, v/v)] is determined to be $-32.4$ kJ mol$^{-1}$

calculation (Supplementary Fig. 6). Molar ratio plot is further acquired by monitoring the absorbance intensity at 480 nm, providing the clear evidence for 1:1 binding stoichiometry between **1** and **3** (Fig. 3b, inset).

In pure methanol, no red-shifted MLCT/LLCT absorbance is observed upon mixing **1** and **3** together (Supplementary Fig. 7a). Hence, we sought to elucidate **1•3** co-assembly behaviors via solvent-dependent UV–Vis measurements, with the systematic variation of water/methanol volume ratio. A sigmoidal denaturation curve is obtained for the **1•3** co-assembly process, by plotting the degree of aggregation ($\alpha_{agg}$) at 480 nm versus MeOH volume fraction ($f$) in water (from 0.10 to 0.35, Fig. 3c). The denaturation curve can be nicely fitted by a previously reported solvent-dependent associating equilibrium model[34], providing $\Delta G$ value (Gibbs free energy gain upon monomer association) of $-38.6$ kJ mol$^{-1}$ in water/methanol (90:10, v/v). Remarkably, the Gibbs free energy release for **1•3** co-assembly process is significantly higher than that of **3** self-assembly process ($\Delta G = -32.4$ kJ mol$^{-1}$, acquired by plotting $\alpha_{agg}$ at 420 nm versus MeOH volume fraction, see Fig. 2d). The dominance of **1•3** hetero-complexation leads to the formation of rod-like micelles (Fig. 3d, and Supplementary Fig. 10), which are distinct from the spherical micelles for the individual rod−coil amphiphile **3** (Fig. 2a). The morphological conversion is primarily ascribed to the hetero π–π stacking interactions between **1** and **3**, which decrease the curvature and thereby release the strain[35].

**Phosphorescent enhancement for complex 1•3.** Intriguingly, although the individual compound **1** or **3** is totally non-emissive

in aqueous environment (quantum yield $\Phi_{em} < 0.1\%$, Fig. 4a), an intense yellow emission signal emerges upon mixing them together ($\lambda_{max} = 568$ nm, together with a shoulder band at 614 nm, $\Phi_{em} = 6.5\%$, Fig. 4a). The large Stokes shift, together with long lifetime in the microsecond range (0.69 µs, Fig. 4b), are indicative of triplet parentage for the emission signal. Gratifyingly, both the emission intensity and lifetime value hardly change between the aerated and deaerated samples of **1•3** (Fig. 4b, and Supplementary Fig. 8). It validates the formation of densely packed domain for co-assembly **1•3**, which protect the triplet excited states from quenching by dissolved molecular oxygen. Noteworthy, the co-assembly of **1•3** in aqueous medium is a prerequisite for the phosphorescent enhancement, since the phenomenon is absent in pure methanol (Supplementary Figs. 7 and 11).

The control experiment is further performed to clarify the emission enhancement mechanism. Specifically, compound **3** is replaced by the commercially available surfactant Brij S20 (Fig. 4b, inset), which is also capable of providing hydrophobic environment for **1** in water. Upon mixing Brij S20 and **1** together (Supplementary Fig. 9), it shows moderate increase of the emission intensity and lifetime ($\lambda_{max} = 619$ nm, $\tau = 0.13$ µs, see Fig. 4a, b). Nevertheless, the phosphorescent quantum yield for complex **1•Brij S20** is 5.7-fold lower than that of complex **1•3** ($\Phi_{em}$: 1.2% for **1•Brij S20** versus 6.5% for **1•3**). As widely known, the emission efficiency is determined by the interplay of radiative and non-radiative processes. For both **1•3** and **1•Brij S20**, they possess the similar radiative decay rates ($k_r$: $0.95 \times 10^5$ s$^{-1}$ for **1•3** versus $0.92 \times 10^5$ s$^{-1}$ for **1•Brij S20**). Evidently, the phosphorescent signals for both complexes arise from the triplet excited states of **1**. On the other

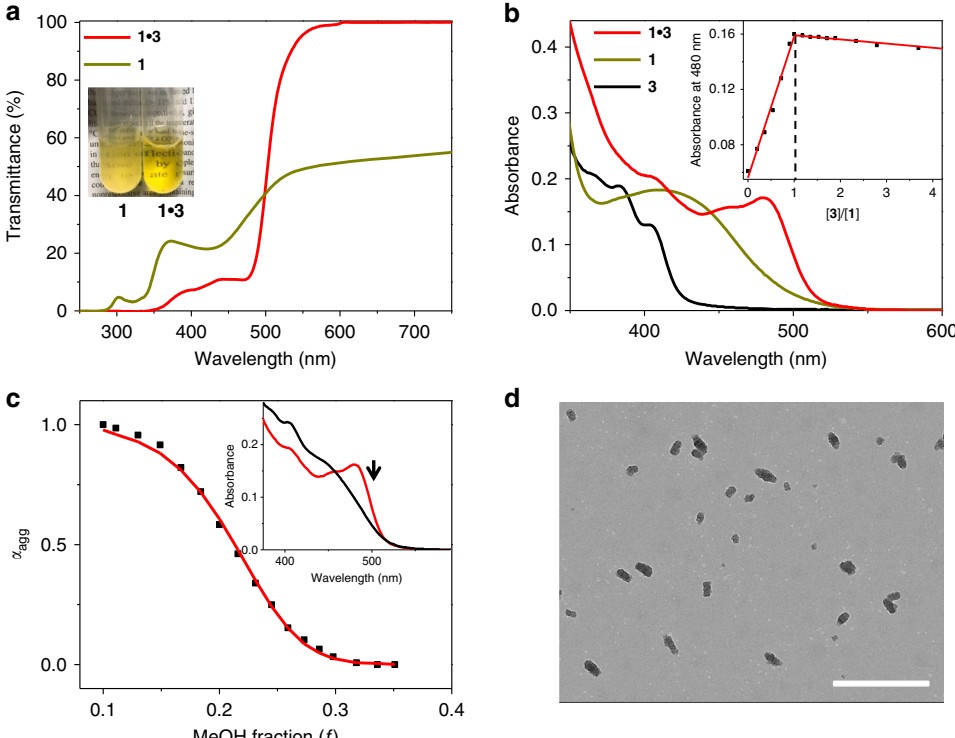

**Fig. 3** Characterization of co-assembled system from **1•3**. **a** Ultraviolet–visible (UV–Vis) transmittance spectra of **1** and **1•3** in pure water ($5.00 \times 10^{-5}$ M for each compound). Inset: images of **1** and **1•3** in pure water ($5.00 \times 10^{-5}$ M for each compound). **b** UV–Vis spectra of **1**, **3**, and complex **1•3** (0.05 mM for each compound) in water/MeOH (90:10, v/v). Inset: molar ratio plot for complex **1•3**, by monitoring the absorption intensity at 480 nm. **c** Degree of aggregation ($\alpha_{agg}$) as a function of MeOH volume fraction monitored at 480 nm. The red line denotes the mathematical fitting of the curve according to the solvent-dependent equilibrium model (see Eqs. (1) and (2) in "Methods"). Inset: solvent-dependent UV–Vis absorption spectra of **1•3** [$5.0 \times 10^{-5}$ M for each compound in water/MeOH (from 90:10 to 65:35, v/v)]. The arrow indicates the spectral changes upon increasing the MeOH fraction. **d** Transmission electron microscopic image recorded on a copper grid by drop-casting complex **1•3** ($5.00 \times 10^{-5}$ M for each compound in water). Scale bar: 500 nm

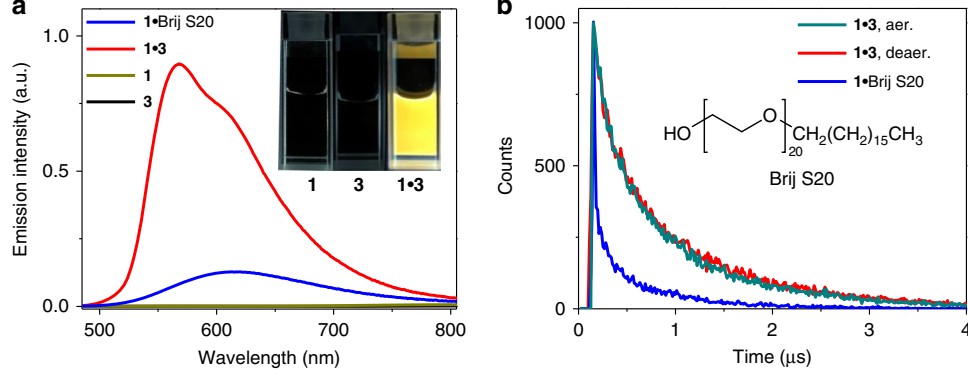

**Fig. 4** Emission characterization of **1•3**. **a** Emission spectra of **1**, **3**, complex **1•3** and complex **1•Brij S20** in water/MeOH (90:10, v/v) (1.00 mM for Brij S20 and 0.05 mM for other compounds). Inset: Emission color images of **1**, **3**, and complex **1•3** under 365 nm ultraviolet lamp. **b** Emission decay traces of the aerated (dark cyan line) and deaerated (red line) samples of complex **1•3**, together with the deaerated sample of complex **1•Brij S20** in water/methanol (90:10, v/v) (1.00 mM for Brij S20 and 0.05 mM for other compounds)

hand, complex **1•3** exhibits five times decrease for the $k_{nr}$ value than that of **1•Brij S20** ($k_{nr}$: $1.38 \times 10^6$ s$^{-1}$ for **1•3** versus $7.60 \times 10^6$ s$^{-1}$ for **1•Brij S20**). The slower non-radiative decay for **1•3**, together with its relatively blue-shifted emission signal indicate the restriction of vibrations and rotations for **1**[25,36]. The phenomenon originates from hetero π-π stacking between **1** and **3**, which is absent in terms of complex **1•Brij S20**.

**Co-assembly behaviors between 2 and 3.** For complex **2•3**, hetero-complexation also prevails in aqueous media, as reflected

by the higher $\Delta G$ value for the **2•3** co-assembly process ($\Delta G = -37.0$ kJ mol$^{-1}$, Supplementary Fig. 13) than that of the **3** self-assembly one ($\Delta G = -32.4$ kJ mol$^{-1}$, Fig. 2d). Consequently, **2•3** displays phosphorescent enhancement phenomenon ($\lambda_{max} = 648$ nm, $\Phi_{em} = 2.4\%$ for **2•3** versus $\Phi_{em} < 0.1\%$ for either **2** or **3**, Fig. 5a, and Supplementary Figs. 12 and 15), thanks to the synergistic participation of shielding and rigidifying effects. As shown in Fig. 5b, the singlet ($S_1$) and triplet ($T_1$) energy levels of **2•3** are determined to be 2.13 and 1.91 eV, respectively. Both values are lower than the $T_1$ energy level of **1•3** (2.18 eV)

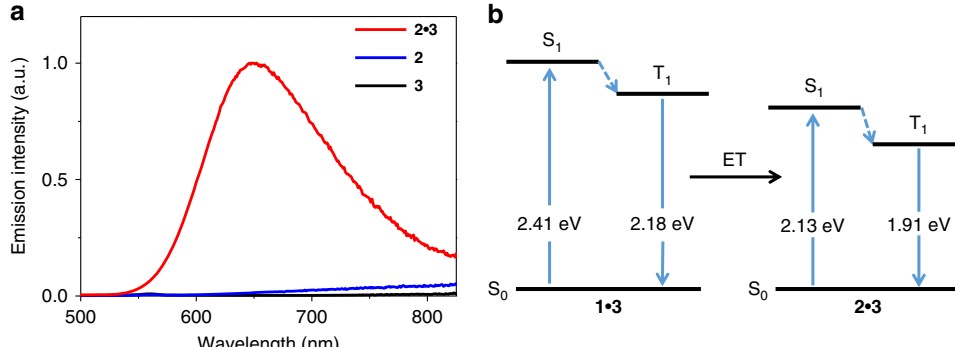

**Fig. 5** Characterization of co-assembled system from **2•3**. **a** Emission spectra of **2**, **3**, and complex **2•3** [$5.00 \times 10^{-5}$ M for each compound in water/methanol (90:10, $v/v$)]. **b** Energy diagram of complexes **1•3** and **2•3**. The triplet energy was determined by the emission maximum (568 and 648 nm for complexes **1•3** and **2•3**, respectively). The singlet energy was determined by the absorption onset on the lower energy side (515 and 580 nm for complexes **1•3** and **2•3**, respectively)

(Fig. 5b). It thus lays the basis for triplet ET upon mixing complexes **1•3** and **2•3** together in aqueous solution.

**Triplet ET for the ternary complex 1•2•3.** Next, the ternary co-assembled complex **1•2•3** is constructed (Fig. 6a). For all of the prepared samples in water/methanol (90:10, $v/v$), the concentration of **3** is equal to the total concentrations of **1** and **2** [denoting as $\mathbf{1•2}_x\mathbf{•3}_{(1+x)}$]. Because of the strong hetero-complexation tendency between **3** and **1** (or **2**), the two platinum(II) phosphors are compartmentalized by **3** to avoid their direct contact. Rod-like micelles are observed for complex $\mathbf{1•2}_{0.1}\mathbf{•3}_{1.1}$ on the basis of TEM measurements (Fig. 6b). The morphology is similar to those of **1•3** (Figs. 3d) and **2•3** (Supplementary Fig. 14), yet distinct from the spherical micelles of the individual compound **3** (Fig. 2a). The phenomenon suggests the co-encapsulation of platinum(II) phosphors **1** and **2** into the hydrophobic interior of **3**.

Light-harvesting capability is further studied for the ternary complex $\mathbf{1•2}_x\mathbf{•3}_{(1+x)}$. Upon gradual increase of **2** as the acceptor unit ($x$: from 0 to 0.1), the emission signal belonging to **1•3** ($\lambda_{\max} = 568$ nm) progressively quenches (Fig. 6c). Meanwhile, the phosphorescence centered at 648 nm strengthens, along with the presence of an iso-emissive point at 610 nm (Fig. 6c). When 3% of **2** is added, the phosphorescent color changes from yellow to red, as directly visualized under 365 nm UV lamp (Fig. 6d, inset). Noteworthy, the red emission signal is significantly weaker for **2•3** in the absence of **1** ($\mathbf{2}_{0.1}\mathbf{•3}_{1.1}$, Fig. 6c). The results unambiguously support the presence of triplet ET from **1** to **2** in the ternary co-assembled system, giving rise to the amplified emission at 648 nm. Additionally, the phosphorescent lifetimes of **1** decline from 0.69 to 0.23 and 0.11 μs for the three prepared samples **1•3**, $\mathbf{1•2}_{0.06}\mathbf{•3}_{1.06}$, and $\mathbf{1•2}_{0.1}\mathbf{•3}_{1.1}$, respectively (Fig. 6d, and Supplementary Fig. 21). The significant decrease of donor lifetime indicates the dominance of non-radiative ET in complex $\mathbf{1•2}_x\mathbf{•3}_{(1+x)}$.

For complexes $\mathbf{1•2}_{0.03}\mathbf{•3}_{1.03}$ and $\mathbf{1•2}_{0.1}\mathbf{•3}_{1.1}$ in water/methanol (90:10, $v/v$), the quantitative ET efficiency ($\Phi_{\mathrm{ET}}$, see Eq. (3) in "Methods") is calculated to be 65.4% and 91.2%, respectively (Fig. 6c). By employing a 1:1 binding isotherm model between triplet donor $(\mathbf{1})_n$ and acceptor **2**, it is found that one acceptor enables the phosphorescence quenching of 65 donor units (Supplementary Fig. 17). It leads to the antenna effect of 15.3 for $\mathbf{1•2}_{0.03}\mathbf{•3}_{1.03}$ and 8.7 for $\mathbf{1•2}_{0.1}\mathbf{•3}_{1.1}$. In addition, the ET rate constant ($k_{\mathrm{ET}}$) in aqueous media is determined to be $2.74 \times 10^6 \mathrm{~s}^{-1}$ and $1.54 \times 10^7 \mathrm{~s}^{-1}$ for $\mathbf{1•2}_{0.03}\mathbf{•3}_{1.03}$ and $\mathbf{1•2}_{0.1}\mathbf{•3}_{1.1}$, respectively (see Eq (4) in "Methods"). Noteworthy, the $k_{\mathrm{ET}}$ value for $\mathbf{1•2}_{0.1}\mathbf{•3}_{1.1}$ is higher than those of the previous triplet ET systems in crystalline

and film phases ($10^5$–$10^6 \mathrm{~s}^{-1}$)[26,37]. It is attributed to the compartmentalization effect rendered by **3**, which organizes platinum phosphors with close spatial proximity yet without direct interaction. The result can be further evidenced by the addition of 10% excessive amount of **3** (Fig. 6e and Supplementary Fig. 18). It increases the distance for the platinum phosphors and consequently leads to three times decrease of the $k_{\mathrm{ET}}$ values.

**ET dynamics of the ternary complex 1•2•3.** For the above ET experiments, they are performed by pre-mixing the stock solution of $\mathbf{1•2}_x\mathbf{•3}_{(1+x)}$ together in methanol, followed by the injection into water (method A in Fig. 7a). In view of the dynamic nature for the ternary co-assembled system, light-funneling efficiency can be influenced by the exchange rates[38]. Taking this into consideration, an alternative fabrication approach has been employed. Briefly, the stock aqueous solutions of **1•3** and **2•3** are separately prepared, followed by mixing of the two solutions together (method B in Fig. 7a). Both methods give rise to the decay of donor emission intensity, as well as strengthening of the acceptor one. However, $\Phi_{\mathrm{ET}}$ value reaches to the plateau after 140 min in method B, which is much slower than that of method A (~20 min) (Fig. 7b). Moreover, $\Phi_{\mathrm{ET}}$ values for method B are determined to be 47.0%, 65.0%, and 74.9% for $\mathbf{1•2}_{0.03}\mathbf{•3}_{1.03}$, $\mathbf{1•2}_{0.06}\mathbf{•3}_{1.06}$, and $\mathbf{1•2}_{0.1}\mathbf{•3}_{1.1}$, respectively. They are comparably lower than the corresponding values obtained in method A (65.4%, 82.0%, and 91.2%, Fig. 7c, and Supplementary Fig. 19). Accordingly, it can be concluded that the exchange dynamics of the ternary co-assembled system exert crucial impact on the triplet ET efficiency.

## Discussion

In summary, an effective strategy has been developed toward phosphorescent emission enhancement and triplet light harvesting in aqueous environment. The design principle is on the basis of the rod−coil-type compartmentalized amphiphile **3**, which provides the synergistic hydrophobic shielding and rigidifying effects for the platinum(II)-based phosphors **1**−**2**. For the ternary co-assembled complex **1•2•3**, the two platinum(II) phosphors are endowed with ordered spatial organization and enhanced emission intensity. Triplet ET can be achieved by varying **1/2** molar ratio, leading to the change of phosphorescent color from yellow to red. Notably, the ET rate constant is $1.54 \times 10^7 \mathrm{~s}^{-1}$ for $\mathbf{1•2}_{0.1}\mathbf{•3}_{1.1}$ in aqueous media, which is one order higher than those of the previous triplet ET systems in crystalline and film phases ($10^5$-$10^6 \mathrm{~s}^{-1}$). Therefore, the compartmentalization

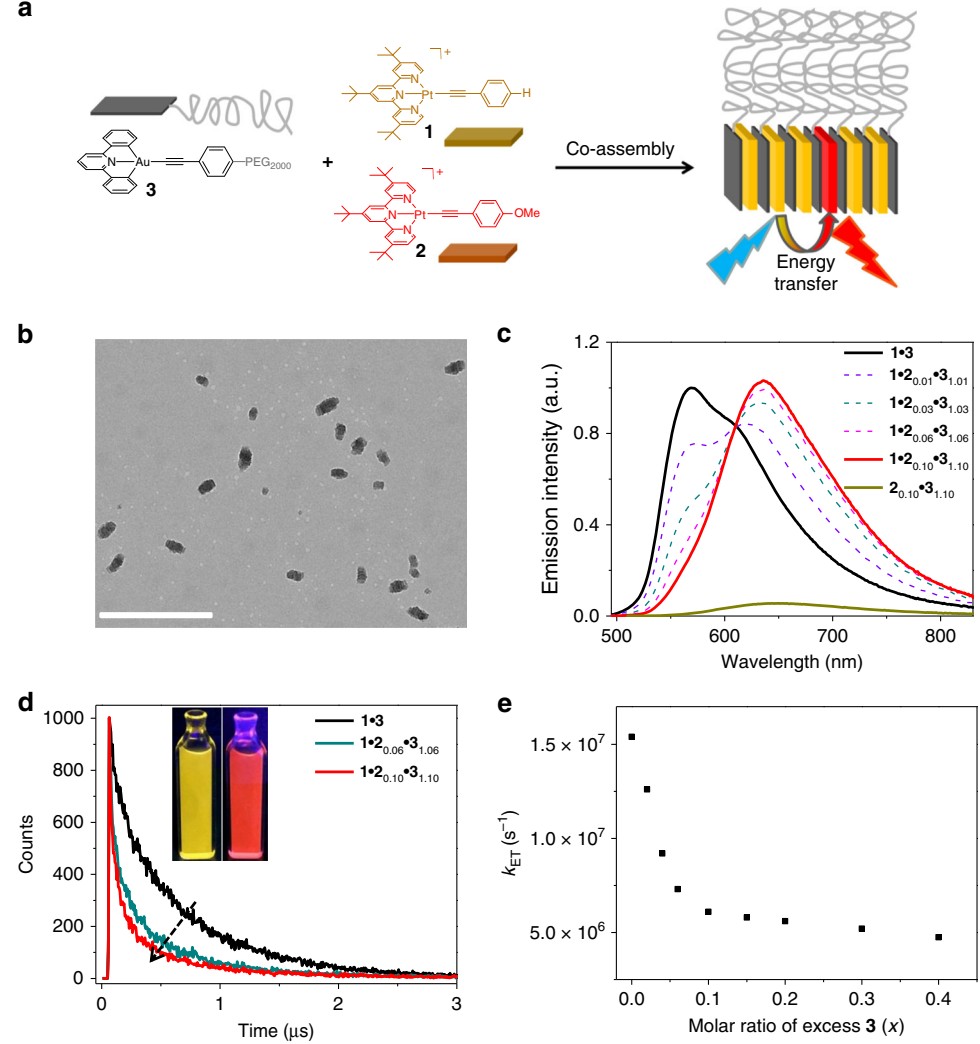

**Fig. 6** Characterization of co-assembled system from **1•2•3**. **a** Schematic representation for the triplet energy transfer of the ternary complex **1•2•3** in aqueous medium. **b** Transmission electron microscopic image recorded on a copper grid by drop-casting complex **1•2**$_{0.1}$**•3**$_{1.1}$ [**1**: $5.00 \times 10^{-5}$ M in water/ MeOH (90:10, v/v)]. Scale bar: 500 nm. **c** Emission spectra and **d** emission decay traces for the ternary complex **1•2**$_x$**•3**$_{(1+x)}$ [**1**: $5.00 \times 10^{-5}$ M in water/ MeOH (90:10, v/v), x varies from 0 to 0.1]. For **d**, the emission wavelength is chosen at 525 nm, considering that **2•3** displays negligible emission at the selected wavelength. Inset of **d**: emission color images of **1•3** (left) and **1•2**$_{0.03}$**•3**$_{1.03}$ (right) under 365 nm ultraviolet lamp. **e** Relationship of the energy-transfer rate constant ($k_{ET}$) value in the ternary complex **1•2**$_{0.1}$**•3**$_{(1.1+x)}$ and the excessive amount of **3** [**1**: $5.00 \times 10^{-5}$ M in water/MeOH (90:10, v/v), x varies from 0 to 0.4]

strategy represents an efficient approach toward phosphorescent light-harvesting materials.

## Methods

**Measurements**. $^1$H nuclear magnetic resonance (NMR) spectra was collected on a Varian Unity INOVA-400 spectrometer with tetramethylsilane as the internal standard. $^{13}$C NMR spectra were recorded on a Varian Unity INOVA-400 spectrometer at 100 MHz. Time-of-flight (TOF) mass spectra were obtained on matrix-assisted laser desorption ionization-time of flight (autoflex speed TOF/TOF, Bruker). UV/Vis spectra were recorded on a UV-1800 Shimadzu spectrometer. Steady-state emission spectra were recorded on FluoroMax-4 spectrofluorometer (Horiba Scientific) and analyzed with an Origin (v8.1) integrated software Fluor-oEssence (v2.2). Solutions for photophysical studies were degassed by purging solvent-saturated N$_2$ for 10 min. The emission quantum yield was determined by using [Ru(bpy)$_3$](PF$_6$)$_2$ ($\Phi_{em} = 6.2\%$ in deaerated CH$_3$CN) as the ref. [39]. Emission lifetime data were acquired with 1 MHz LED laser with the excitation peak at 455 nm (NanoLED-455) and analyzed with DataStation v6.6 (Horiba Scientific). TEM experiments were performed on Tecnai G2 Spirit BioTWIN electron microscope (acceleration voltage: 120 kV). The hydrodynamic size distributions were characterized using DLS (Malvern Zetasizer Nano ZS90). DFT computations were performed by utilizing the Gaussian 09 D.01 software package. During the optimization process, Lanl2dz core potential was chosen to describe Pt and Au

atoms, while all other elements were described by using ωb97xd and 6-31G as dispersion corrected exchange functional and basis set, respectively.

**Mathematical curve fitting of the assembling process**. To acquire the detailed thermodynamic parameters for the solvent-dependent self-/co-assembly process, a sigmoidal denaturation curve is obtained upon plotting the fraction of aggregates ($\alpha_{agg}$) versus the MeOH volume fraction ($f$) in water.

$$\alpha_{agg} = \frac{A(\lambda) - A_M}{A_{agg} - A_M} \quad (1)$$

In this equation, $A(\lambda)$ is the absorbance at a given MeOH volume fraction. $A_M$ is the absorbance at the high MeOH volume fraction, corresponding to the mono-meric state. $A_{agg}$ is the absorbance at the low MeOH volume fraction, corresponding to the fully aggregated state. The curve can be further fitted by an equilibrium model adapted to solvent-dependent association[36], which is described by Eq. (2):

$$\Delta G_f = \Delta G_0 + m \times f \quad (2)$$

In this equation, $\Delta G_0$ represents the Gibbs free energy gain upon monomer association in pure water. The dependence of Gibbs free energy ($\Delta G_f$) on the solvent ($f$) is described by the $m$ value.

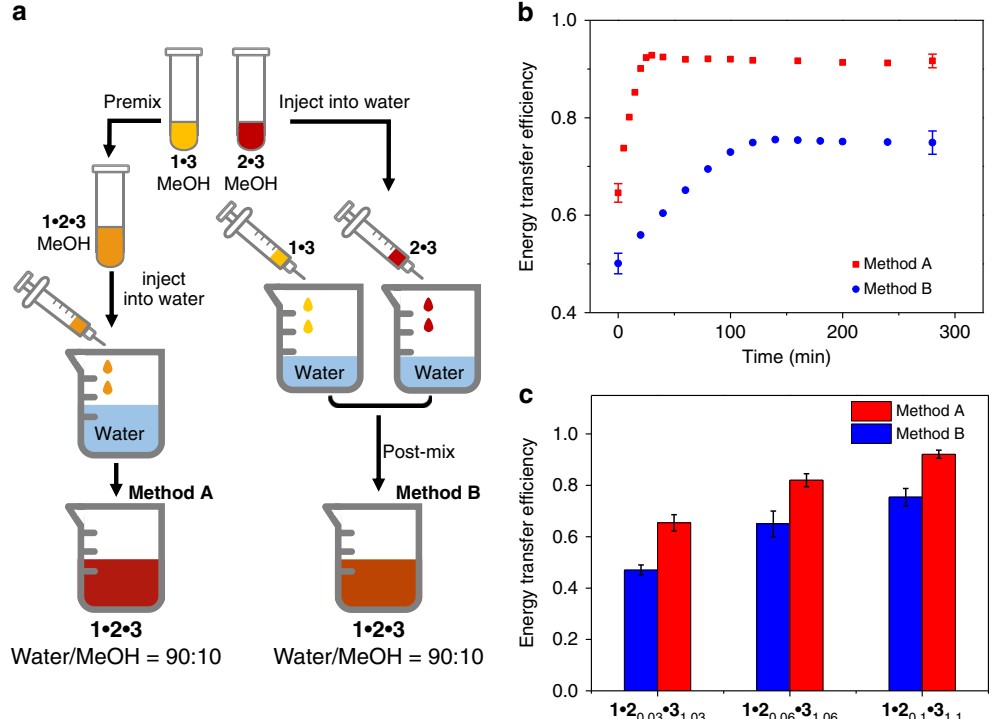

**Fig. 7** Characterization of energy transfer dynamics of **1•2•3**. **a** Schematic illustration of two preparation methods toward the ternary complex **1•2**$_x$**•3**$_{(1+x)}$. **b** Time dependence of energy transfer efficiency for complex **1•2**$_{0.1}$**•3**$_{1.1}$. **c** Energy transfer efficiency for complexes **1•2**$_{0.03}$**•3**$_{1.03}$, **1•2**$_{0.06}$**•3**$_{1.06}$, and **1•2**$_{0.1}$**•3**$_{1.1}$, by employing the two different preparation methods [**1**: $5.00 \times 10^{-5}$ M in water/MeOH (90:10, $v/v$)]. All error bars represent s.e.m. from triplicate independent measurements

**Determination of the ET efficiency.** ET efficiency ($\Phi_{ET}$) is calculated according to Eq. (3).

$$\Phi_{ET} = 1 - I_{DA}/I_D \qquad (3)$$

In the equation, $I_{DA}$ and $I_D$ are the emission intensity of ET donor with and without the presence of ET acceptor, respectively.

ET rate constant ($k_{ET}$) is calculated according to Eq. (4).

$$\Phi_{ET} = k_{ET}/(k_{ET} + \tau_D^{-1}) \qquad (4)$$

In the equation, $\tau_D$ is the phosphorescent lifetime of **1•3** (served as the ET donor).

**Calculation of the donor number quenched by acceptor.** In the ternary system **1•2**$_x$**•3**$_{(1+x)}$, the majority of emission for the donor (complex **1•3**) was quenched by a few percentage of acceptor (complex **2•3**). If $n$ donors are assumed to be quenched by one acceptor, we could treat (D)$_n$ as one unit to model the quenching process via the 1: 1 binding isotherm. This model combines the direct quenching and energy-migration-assisted ET pathways, which is expressed by Eq. (5),

$$(D)_n + A = (D)_n \bullet A \qquad (5)$$

The value of $n$ is further obtained in Eq. (6), by nonlinear least-squares fittings of emission intensity of donor versus the acceptor concentration.

$$I_D = I_0 + ((I_{lim} - I_0)/(2 \times C_0)) \times (C_0 + C_A + (1/K_a) - ((C_0 + C_A + (1/K_a)) \wedge 2 - 4 \times C_A \times C_0) \wedge (1/2)) \qquad (6)$$

In this equation, $I_D$ is the emission intensity of the ET donor, and $I_0$ is the emission intensity of donor in the absence of acceptor. $I_{lim}$ is the emission intensity of the donor, which has fully complexed with the acceptor. $C_0$ is the concentration of (D)$_n$, while $C_A$ is the concentration of acceptor. $K_a$ is the association constant between (D)$_n$ and acceptor (A).

## Data availability

The data that support the findings of this study are available from the corresponding author on request.

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

## Acknowledgements

We acknowledge the funding support from National Natural Science Foundation of China (21871245, 21674106), CAS Youth Innovation Promotion Association (2015365), and the Fundamental Research Funds for the Central Universities (WK3450000004).

## Author contributions

Z.L. and F.W. conceived the idea for this project. Z.L. performed the experiments, analyzed the data, and produced the artwork under the direction of F.W. Y.H. contributed to the theoretical calculations. All authors contributed to the manuscript preparation.

## Additional information

**Competing interests:** The authors declare no competing interests.

