## [Peer Review File · Nature Communications]

REVIEWERS' COMMENTS:

Reviewer #1 (Remarks to the Author):

Wang and co-workers have demonstrated Phosphorescent Emission Enhancement and
Triplet Energy Transfer in Aqueous Medium using a supramolecular
compartmentalization approach. Achieving ambient phosphorene is important from an
application perspective and the compartmentalization strategy presented here is indeed an
attractive approach in this direction. I would recommend the acceptance of this work after
following revisions.

1. Can the authors elaborate the mechanism of quenching of Pt-complexes (1 and 2) in
water? And does the deoxygenation of aqueous solutions help improving/detecting the
phosphorescence?

2. Main text, page 4. There is a typo error, its molecule 3, not 1.

3. Reference 6 is wrongly written.

4. The authors termed "triplet energy transfer (ET)" throughout the manuscript. However
never designated whether it is FRET or a Dexter process. It is important to know,
whether triplet (from molecule 1) to singlet (molecule 2) FRET is occurring in this
system. We know Dexter requires a very close distance between two dyes (D and A)
which is not the case for FRET. Thus understanding this process would help in designing
molecular system in a better way

5. The authors should provide data of energy transfer experiments at higher concentration
regime. It seems like, a Dexter process is occurring here (distance between D and A is
below 0.5 nm), thus it is important to know the presence of a triplet-triplet annihilation
process.

6. It will be interesting to see whether the solution state self-assembly is maintained in the
solid state, which is particularly important for device applications. Can the authors
provide any data on this?

Reviewer #2 (Remarks to the Author):

In this manuscript, Wang and coworkers presented a novel and effective strategy to
design and construct the donor-acceptor phosphorescent systems to realize triplet energy
transfer. They prepared a rod-coil amphiphile as the compartmentalized agent.
Moreover, they employed the platinum(II) phosphors with synergistic rigidifying and
hydrophobic shielding effects to enhance the phosphorescent emission in aqueous
environment. Impressively, they revealed that the donor-acceptor platinum(II) phosphors
feature an ordered spatial organization in the multi-component co-assembled system and
the efficient light-harvesting can be achieved in aqueous medium with the unprecedented
triplet energy-transfer rate constant by varying donor/acceptor molar ratio. It should be
noted that the construction of artificial energy transfer system has been always an

attractive topic within chemistry and materials science. There is no doubt that the authors
presented a very important advance in this field, which should receive much broad
general interest. The manuscript is well-organized and the conclusion is very solid. So I
strongly recommend it to be published in Nature Communications after very minor
revision.

1. It is well known that phosphorescence can be quenched by oxygen. I am wondering if
the authors have tried to conduct the photo physical measurements under de-oxygenated
or oxygenated condition. It will be very helpful to get the insight into the photo physical
property of the resultant assemblies.

2. It is very impressive to learn that an intense yellow emission signal emerges upon
mixing individual compound 1 and 3 together in this study. Moreover, the authors
presented the uniformed morphology of aggregation of co-polymers in water. So if it
allows, the confocal laser scanning microscope is a very useful technique to demonstrate
such phenomena.

3. It will be very helpful for readers to learn the mechanism of triplet energy transfer
process if the authors can provide a cartoon figure to show the mechanism in the main
text.

4. Some recently reports on supramolecular D–A systems, in particular those contain Pt,
are encouraged to be cited in the revised version, which will be very helpful for readers to
learn this area (J. Am. Chem. Soc., 2017, 139, 9459; J. Am. Chem. Soc. 2016, 138, 738).

Reviewer #3 (Remarks to the Author):

This manuscript reports on increased rates of triplet energy transfer to enhance
phosphorescence by co-assembly of donor-acceptor Pt complexes in conjunction with Au
complexes. The supramolecular assembly into rod-like structures enables fast triplet
energy transfer and enhanced phosphorescence lifetime of the acceptor due to shielding
from quenching by molecular oxygen. The work seems to be conducted carefully using a
variety of experimental techniques. I recommend to accept this manuscript after the
detailed comments below have been addressed:

1. Page 1, line 3 from bottom: “unprecedented triplet energy-transfer rate constant ($\sim 10^7$
S^{-1}).” Although the rate constant of triplet energy transfer is very fast, I would not say
that it is “unprecedented”. Faster triplet energy transfer rate constants have been routinely
achieved in intramolecular triplet energy transfer for organic chromophores.
“unprecedented” should be replaced with “very high” or a similar, non-exaggerating
phrase.

2. Page 3, line 5: “narcissistic homo-complexation ones” The use of “narcissistic” in this
phrase is probably not recommended.

- 3. Page 13, line 7 from bottom: “[Ru(bpy)₃](PF₆)₂ ($\Phi_{em} = 0.062$ in deaerated CH₃CN)”
A literature reference should be provided for this quantum yield.
- 4. Page S4, Figure caption for Figure S4: “The critical micelle concentration (CMC) of 3
is determined to be 6.70×10^{-6} M.” A 3-diggit accuracy for the CMC is probably not
justified with the used experimental technique. “ 6.7×10^{-6} M” is more realistic.
- 5. Page S6, Figure S8: “b) life decay of 1” change to: “b) emission decay traces of 1”

**Reviewer #1:**

*1) Can the authors elaborate the mechanism of quenching of Pt-complexes (1 and 2) in*
*water? And does the deoxygenation of aqueous solutions help improving/detecting the*
*phosphorescence?*

**Response:** Since the platinum(II) complexes **1** and **2** are not soluble in pure water,
water/MeOH (90 : 10, v/v) is employed as an alternative to perform the emission
measurements (Fig. R1a). As widely known, the monomeric state dominates for the
platinum(II) complexes in chlorinated solvents, while aggregation tends to occur in
highly polar solvent (F. Würthner *et al. Chem. Soc. Rev.*, **2009**, 38, 564–584). For **1**, the
emission intensity declines upon increasing the solvent polarity [Φ_{em} : from 8.3% in
CHCl₃, 0.46% in MeOH to < 0.1% in water/MeOH (90 : 10, v/v)] (Fig. R1b). No
emission enhancement is observed for **1** in water/MeOH (90 : 10, v/v) under deaerated
condition ($\Phi_{em} < 0.1\%$) (Fig. R1c). The similar emission behavior is observed for **2** (Fig.
R1d). Hence, self-aggregation caused quenching (ACQ) is the main reason for the
emission intensity decrease of platinum(II) complexes. The conclusion can be further
supported by the decrease of photoluminescence quantum yield for both **1** and **2** in the
solid state ($\Phi_{em} < 0.1\%$, Fig. R1c–d, inset).

**Figure R1.** a) UV–Vis transmittance spectra of **1** (5.00×10^{-5} M) in pure water (red line)
 and water/MeOH (90 : 10, v/v) (black line). b) Emission spectra ($\lambda_{\text{ex}} = 470$ nm) of
 platinum(II) phosphor **1** (5.00×10^{-5} M) in different solvents. c) Emission spectra of the
 aerated (dark cyan line) and deaerated (red line) samples of **1** in water/MeOH (90 : 10,
 v/v). d) Emission spectra of the aerated (dark cyan line) and deaerated (red line) samples
 of **2** in water/MeOH (90 : 10, v/v). Inset of c–d): emission color images of c) **1** and d) **2** in
 the solid state under 365 nm UV lamp. The absolute emission quantum yields are
 measured on FluoroMax-4 spectrofluorometer with an integrating sphere.

*2) Main text, page 4. There is a typo error, its molecule 3, not 1.*

**Response:** We really appreciate that the reviewer pointed this out. We have revised the
 error in the mentioned place.

*3) Reference 6 is wrongly written.*

**Response:** We really appreciate that the reviewer pointed this out. We have revised the
error in the mentioned place.

*4) The authors termed "triplet energy transfer (ET)" throughout the manuscript. However*
*never designated whether it is FRET or a Dexter process. It is important to know,*
*whether triplet (from molecule 1) to singlet (molecule 2) FRET is occurring in this system.*
*We know Dexter requires a very close distance between two dyes (D and A) which is not*
*the case for FRET. Thus understanding this process would help in designing molecular*
*system in a better way.*

**Response:** As stated in Fig. 5b of the main text, the donor species features a triplet state
energy (T_1) of 2.18 eV, which is higher than that of the singlet (S_1 , 2.13 eV) and triplet
(T_1 , 1.91 eV) level of the acceptor species. It suggests that triplet energy transfer can take
place by either T_1 - S_1 Förster or T_1 - T_1 Dexter process for the ternary system **1•2•3**.

Moreover, one acceptor enables the phosphorescence quenching of 65 donor units in
the ternary system **1•2•3**, by calculating the quantitative energy-transfer efficiency (see
Supplementary Fig. 17). On this basis, we can establish a simplified packing model, in
which the ternary co-assembled system **1•2•3** are organized in a rectangle box (Fig. R2,
the numbers of **3**, donor **1** and acceptor **2** are 66, 65, 1, respectively, while the acceptor
moiety is placed in the middle of the rectangle). In combination with the molecular size
information acquired *via* DFT calculation, it can be concluded that the longest distance
(R_0) between donor and acceptor units is greater than 25 Å (Fig. R2). Since the rate of
triplet energy transfer by electron exchange falls exponentially with distance, energy
transfer in the current ternary co-assembled system **1•2•3** can not be exclusively ascribed
to the short-range Dexter process (typically within 10 Å). In other word, donor-acceptor
Förster process, together with exciton energy migration within the donor species, could
also participate in the energy transfer process. Similar behaviors are also encountered for
the previously reported triplet energy transfer systems in the gel and crystalline states
(Zhong, Y.-W. *et al. J. Am. Chem. Soc.*, **2018**, *140*, 4269–4278; Schanze, K. S. *et al. J.*
*Am. Chem. Soc.*, **2008**, *130*, 2535–2545). Hence, more in-depth experimental and
theoretical studies are required to distinguish the exact contribution from Förster and
Dexter processes, which will be investigated in our future work.

**Figure R2.** Schematic representation of the simplified packing model of the ternary
 complex **1•2•3**.

*5) The authors should provide data of energy transfer experiments at higher*
 *concentration regime. It seems like, a Dexter process is occurring here (distance between*
 *D and A is below 0.5 nm), thus it is important to know the presence of a triplet-triplet*
 *annihilation process. And 6) It will be interesting to see whether the solution state self-*
 *assembly is maintained in the solid state, which is particularly important for device*
 *applications. Can the authors provide any data on this?*

**Response:** The two questions have some similarity, since the solid state can be regarded
 as the extremely concentrated medium for donor–acceptor energy transfer. In this respect,
 we sought to investigate triplet energy transfer behaviors of the ternary co-assembled
 system **1•2•3** in the solid state.

As an initial step, co-assembly between the platinum compound **1** and the
 compartmentalized agent **3** is studied. As can be seen, **1** displays very weak emission
 intensity in the solid state, because of the self-aggregation caused quenching effect (Fig.
 R3a–b). In the meantime, **3** is almost non-emissive, owing to the presence of thermally-
 accessible *d-d* or LLCT (ligand-to-ligand charge transfer) states for the
 [Au(III)(C[^]N[^]C)(C≡C–R)] unit (Fig. R3a–b). Interestingly, upon mixing the equimolar
 amount of **1** and **3** together and grinding them for 5 min, an intense yellow emission

signal appears (the peak maximum is centered at 577 nm, while the lifetime is 0.62 μ s,
 see Fig. R3a–b). The emission enhancement phenomenon of complex **1•3** in the solid
 state is consistent with that in the aqueous medium [$\lambda_{\text{max}} = 568$ nm, $\tau = 0.69$ μ s in
 water/MeOH (90 : 10, v/v)]. Similar emission enhancement is visualized for complex **2•3**
 in the solid state ($\lambda_{\text{max}} = 641$ nm, $\tau = 0.42$ μ s, see Fig. R3c–d). Hence, it is evident that
 phosphorescent emission enhancement for the two-component co-assembled systems not
 only exists in the aqueous state, but maintains in the solid state.

 **Figure R3.** a) Emission spectra of **1**, **3**, and complex **1•3** in solid state. b) Emission decay
 trace of complex **1•3** in solid state. Inset: emission color images of **1**, **3** and complex **1•3**
 in the solid state under 365 nm UV lamp. c) Emission spectra of **2**, **3**, and complex **2•3** in
 solid state. d) Emission decay trace of complex **2•3** in solid state. Inset: emission color
 images of **2**, **3** and complex **2•3** in the solid state under 365 nm UV lamp.

On this basis, we turned to investigate triplet energy transfer behaviors of the ternary
 complex **1•2•3** in the solid state. To guarantee ideal donor–acceptor spatial organization
 in the ternary co-assembled system, one drop of methanol is added to the mixture of **1•3**

and 10% amount of **2•3**. After grinding for 5 minutes, the mixed sample is further dried
 in vacuum to remove the solvent. As can be seen, the phosphorescent color changes from
 yellow of **1•3** to red of **1•2_{0.1}•3_{1.1}** (Fig. R4a–b). Moreover, the decay lifetime shortens
 from 0.62 μs of **1•3** to 0.10 μs of **1•2_{0.1}•3_{1.1}** (Fig. R4c). The result unambiguously
 supports the involvement of triplet energy transfer in the ternary co-assembled system.
 The energy transfer efficiency is determined to be 84% for **1•2_{0.1}•3_{1.1}** in the solid state,
 which is comparable to the same sample in aqueous environment [91% in water/MeOH
 (90 : 10, v/v)]. Overall, phosphorescence enhancement and triplet energy transfer
 maintain for the ternary complex **1•2•3** in the solid state. Notably, neither quenching nor
 up-conversion of the emission signal is observed. It suggests the absence of triplet–triplet
 annihilation in the solid state, primarily ascribing to the compartmentalization effect
 provided by **3**.

**Figure R4.** a) Emission color images of **1•2_{0.1}•3_{1.1}** under 365 nm UV lamp. b) Emission
 spectra, and c) emission decay traces of complex **1•3** and **1•2_{0.1}•3_{1.1}** in the solid state. For
 lifetime measurements, the emission wavelength is chosen at 525 nm, considering that
 **2•3** displays negligible emission at the selected wavelength.

**Reviewer #2:**

*1) It is well known that phosphorescence can be quenched by oxygen. I am wondering if*
*the authors have tried to conduct the photo physical measurements under de-oxygenated*
*or oxygenated condition. It will be very helpful to get the insight into the photo physical*
*property of the resultant assemblies.*

**Response:** As widely documented, molecular oxygen tends to interact with the occupied
d_z^2 orbital of platinum(II) complexes, and thereby induces the phosphorescent emission
quenching (L. De Cola. *et al. Chem. Soc. Rev.*, **2014**, 43, 4144–4166). For the current co-
assembled system such as complex **1•3**, both the emission intensity and lifetime hardly
change between the aerated ($\Phi_{em} = 6.1\%$, $\tau_0 = 0.67 \mu s$) and deaerated ($\Phi_{em} = 6.5\%$, $\tau_0 =$
$0.69 \mu s$) samples in water/methanol (90 : 10, v/v) (Fig. R5). The phenomena suggest the
formation of densely packed domain for co-assembly **1•3**, which excludes the possibility
for interaction between oxygen and platinum(II) species. Accordingly, the co-assembled
system presented in the current work provides a plausible strategy to protect the triplet
excited states from quenching by dissolved molecular oxygen.

**Figure R5.** a) Emission spectra ($\lambda_{ex} = 470 \text{ nm}$), and b) emission decay traces of complex
**1•3** [$5.00 \times 10^{-5} \text{ M}$, water/methanol (90 : 10, v/v)] in the aerated (black line) and
deaerated (red line) conditions.

*2) It is very impressive to learn that an intense yellow emission signal emerges upon*
*mixing individual compound 1 and 3 together in this study. Moreover, the authors*
*presented the uniformed morphology of aggregation of co-polymers in water. So if it*

*allows, the confocal laser scanning microscope is a very useful technique to demonstrate*
*such phenomena.*

**Response:** Based on the reviewer's comment, we have tried to characterize the co-
assembled structures by means of confocal laser scanning microscopy. Unfortunately, no
reliable result can be obtained. It could be ascribed to the relatively small sizes of the co-
assembled systems [30–45 nm for **1•3** and **2•3** in in water/MeOH (90 : 10, v/v)], which
are below the lower limit of confocal laser scanning microscopy.

*3) It will be very helpful for readers to learn the mechanism of triplet energy transfer*
*process if the authors can provide a cartoon figure to show the mechanism in the main*
*text.*

**Response:** Based on the reviewer's suggestion, we have added a simplified cartoon
figure (Fig. 6a in the revised main text) to clarify the mechanism of triplet energy transfer
process.

*4) Some recently reports on supramolecular D–A systems, in particular those contain Pt,*
*are encouraged to be cited in the revised version, which will be very helpful for readers*
*to learn this area (J. Am. Chem. Soc., 2017, 139, 9459; J. Am. Chem. Soc. 2016, 138,*
*738).*

**Response:** According to the reviewer's suggestion, we have added the mentioned
literatures in the reference part (Ref. 12 and 19).

**Reviewer #3:**

*1) Page 1, line 3 from bottom: “unprecedented triplet energy-transfer rate constant ($\sim 10^7$*
*S^{-1})." Although the rate constant of triplet energy transfer is very fast, I would not say*
*that it is “unprecedented”. Faster triplet energy transfer rate constants have been*
*routinely achieved in intramolecular triplet energy transfer for organic chromophores.*
*“unprecedented” should be replaced with “very high” or a similar, non-exaggerating*
*phrase.”*

**Response:** We agree with the reviewer's suggestion. Accordingly, we have modified the
description in the revised manuscript.

*2) Page 3, line 5: "narcissistic homo-complexation ones" The use of "narcissistic" in*
*this phrase is probably not recommended.*

**Response:** According to the reviewer's suggestion, we have removed the word
"narcissistic" in the revised manuscript.

*3) Page 13, line 7 from bottom: "[Ru(bpy)₃](PF₆)₂ ($\Phi_{em} = 0.062$ in deaerated CH₃CN)"*
*A literature reference should be provided for this quantum yield.*

**Response:** According to the reviewer's suggestion, we have added a related literature in
the reference part (Ref. 39).

*4) Page S4, Figure caption for Figure S4: "The critical micelle concentration (CMC) of*
*3 is determined to be 6.70×10^{-6} M." A 3-diggit accuracy for the CMC is probably not*
*justified with the used experimental technique. " 6.7×10^{-6} M" is more realistic.*

**Response:** We really appreciate that the reviewer pointed this out. We have revised the
error.

*5) Page S6, Figure S8: "b) life decay of I" change to: "b) emission decay traces of I".*

**Response:** According to the reviewer's suggestion, we have revised the error.

Some of the data, results, and discussion are added in the main text and supporting
information (see the revised manuscript with highlight). We are hopeful that it now meets
your standards for acceptance. Thanks!

REVIEWERS' COMMENTS:

Reviewer #1 (Remarks to the Author):

Authors have done additional experiments to satisfy my concerns and I think the revised
manuscript is suitable for publication in Nature Communications.

Reviewer #2 (Remarks to the Author):

This is a revised manuscript resubmitted by Wang and coworkers, in which they
presented a novel and effective strategy to design and construct the donor–acceptor
phosphorescent systems to realize triplet energy transfer. I have checked the revised
version very carefully. I can tell that the authors have revised the manuscript very
carefully according to the reviewers' suggestion. Now the quality of the revised draft has
a large step compared to the original one. So the current version should be accepted by
Nature Communications.